# Comparative analysis of experimental and numerical investigation on multiple projectile impact of AA5083 friction stir welded targets

S. Balaji[1], S. Dharani Kumar[2], U. Magarajan[3], S. RameshBabu[4], S. Ganeshkumar[5], Shubham Sharma[6,7]*, Shaimaa A. M. Abdelmohsen[8], Indranil Saha[9], Sayed M. Eldin[10]*

1 Material Science and Technology, Technical University of Bergakademie Freiberg, Freiberg, Germany, 2 KPR Institute of Engineering and Technology, Centre for Machining and Material Testing, Coimbatore, India, 3 Bharath Institute of Higher Education and Research, Department of Mechanical Engineering, Chennai, Tamil Nadu, India, 4 KPR Institute of Engineering and Technology, Department of Mechanical Engineering, Coimbatore, India, 5 Department of Mechanical Engineering, Sri Eshwar College of Engineering, Coimbatore, India, 6 Mechanical Engineering Department, University Center for Research and Development, Chandigarh University, Mohali, Punjab, India, 7 School of Mechanical and Automotive Engineering, Qingdao University of Technology, 266520, Qingdao, China, 8 Department of Physics, College of Science, Princess Nourah Bint Abdulrahman University, Riyadh, Saudi Arabia, 9 Department of Physics, GLA University, Mathura, U.P., India, 10 Faculty of Engineering, Center for Research, Future University in Egypt, New Cairo, Egypt

* shubham543sharma@gmail.com, shubhamsharmacsirclri@gmail.com (SS); elsayed.tageldin@fue.edu.eg (SME)

**Data Availability Statement:** All relevant data are within the paper.

## Abstract

This research aims to investigate the ballistic resistance of base material (BM)and "Friction Stir Welded (FSW)", AA5083 aluminum alloy. The primary objective was to build a finite element model to predict kinetic energy absorption and target deformation under single and multiple projectile impact conditions. This study employed 7.62mm Hard Steel Core (HSC) projectiles produced from Steel 4340. The target was analyzed using commercially available Abaqus Explicit software for Finite Element Analysis. It was noticed that the generation of kinetic energy and surface residual velocity increases as the number of projectile strikes increases. In addition, the experimental ballistic test was conducted to validate the numerical results. Using the analytical Recht-Ipson model, each target's experimental residual velocity was determined. It was determined that weldments perform less well (30%) as compared to BM targets. Occurrence of plastic deformation during welding causes reduction in ballistic performance of weldments. For both the computational and experimental approaches, a correlation between residual velocities was found. The plastic deformations with ductile hole formation were observed in all the cases.

## 1. Introduction

Early in the 1950s, ballistic research was carried out for military purposes [1]. Ballistics is a discipline of mechanics that concentrates on the projectile in the weapon barrel, the projectile's behaviour during flight, and the effects on both the projectile and the target after impact [2].

**Funding:** The author(s) received no specific funding for this work.

**Competing interests:** The authors have declared that no competing interests exist.

Typically, it is divided into three categories: interior, exterior, and terminal ballistics [2]. As a result of the projectile's effect on the target, scientists are concerned with terminal ballistics research in order to create a more powerful and effective weapon against enemies. Over the ages, protection against ballistic impact is the major problem due to abrupt failure of existing materials. The armoured vehicle's weight with good ballistic protection against bullet impacts is a continuing goal of the industry. Therefore, the armour's performance should be such that the crew is adequately shielded from the high-velocity bullet. High impacts are typically brought on by the projectile's low mass and high velocity. Aluminum alloys are frequently utilized for their light weight and accessibility. The projectile impacts on AA5083 are researched and investigated in the current work. It is preferred for its affordability, strong mechanical strength with improved corrosion resistance, good ballistic characteristics, and weldability. AA5083 alloy is used for the manufacturing of military vehicle parts such as shells, turrets, drive trains, and weapon sights [3]. A key component in the creation of terminal ballistics is AA5083. It exhibits enhanced protection and significant mobility. That is why the defense industry uses aluminium for combat vehicles. The 5083 grade's composition exhibits increased rigidity to support the weight without breaking or deforming. For various boundary conditions, projectile shapes, and attack angles, numerous researchers had calculated the projectile impact of AA5083 target and armour joints [1–4]. From a single impact, numerous minor consequences were produced by target deformations acting as a failure mechanism. The strength model, failure model, and erosion model are all improved by the numerical investigation [1]. The ballistic limit velocity of the following materials—AA 7075-T6, AA 5083-H116, titanium, and Kevlar 149—was calculated numerically by Suresh et al. [2]. They attempted to expand the use of the honeycomb in order to get the best ballistic limit velocity. ANSYS Workbench software was used for finite element analysis (FEA). As a result, the lower the reported limit velocity for the AA5083-H116 target, the higher the deformation. According to the MIL-DTL-46027J (MR) standard, Gooch et al. [5] found the ballistic resistance for the aluminium alloy AA5083. The authors established an "empirical correction factor" in order to quantify the effect of the AP projectile's jacket, which encases its steel core [6]. The paper further discussed about the angular targets and its effects. Badour et al. [7] investigated the welding of AA 5083 with a dissimilar alloy using the Coupled Eulerian-Lagrangian approach and finite element modelling. The thermal effects of the weld region are explained in terms of the joint's hardness and quality. It described the use of featureless and featured pins for determining the superior weld quality. Jung et al. [8] studied the ballistic behaviour of AA5083-H112 plates using 7.62 mm projectile. They found that a ductile hole with a strain hardening effect was the main cause of the failure mode, which led to it. Srikanth et al. [9] studied the "ballistic resistance" of "cold-metal transfer welded joints" of "AA5083", and "AA6061 thin sheets". Saravanakumar et al. [10] studied the ballistic behaviour of underwater friction-stir welded AA5083 targets. The joint produced by hexagonal pin profile shows absence of fragmentation and less adiabatic shear band lines. Praveen et al. [11] studied the ballistic behaviour of thick FSW AA7075-T651 targets. They found that the depth of penetration of stir zone was less than base material. Thin plate experimental results can be used as a standard for comparing ABAQUS simulation results. Flores-Johnson et al. [12] found that impact behavior of welded plates can be predicted accurately by using the shear-failure and Johnson cook material model.

The ballistic behaviour of GMAW and friction stir welded AA5083 thin sheets was investigated by Dannemann et al. [13]. They found the less ballistic performance and more damage was observed in the GMAW joints as compared to FSW joints. From this, it was noticed FSW joining technique better than GMAW. The friction Stir welded, rolled, and cast AZ31B Mg alloy has somewhat better ballistic performance than the base material (BM) [14,15]. Ballistic performance of FSW welded aluminum alloy was better compare to the GMAW [16]. Cho

et al. [16] found a similar type of study in the friction stir welded AA7XXX series aluminium-alloy. However, they evaluated the $V_{50}$ of the welded and heat-affected regions by using indentation testing methods. The effect of ceramics on the ballistic behaviour of FSP AA6061 aluminium alloy plates was examined by Magarajan et al. [17]. The ballistic resistance of the plate was increased due to the surface microhardness. The multiple hits of the projectile caused more cracks in the FSP AA6061 targets compared to a single hit [18]. Over all, from the literature, it was noticed that FSW and FSP are the better manufacturing processes for fabricating defence structural parts. Yu et al. [19] created a "thermo-mechanical model" for the FSW process using the ABAQUS environment. Iqbal et al. [20] assessed the "ballistic performance" of "monolithic", and "multi-layered mild steel targets" using "ABAQUS software". They arrived to the conclusion that the experimental data and the ballistic numerical calculations were consistent. The element size and aspect ratio are two important criteria in the ABAQUS study [21]. Damage criteria are also crucial for obtaining reliable numerical ballistic results [22]. It was discovered that the correlation between experimental and numerical ballistic impact analysis of two distinct heat treated AA7075 aluminium alloy targets was stronger [23–25]. The vast majority of research on AA5083 targets [8,9] has been conducted using thin sheets and single hits. In reality, military vehicles are built of plates and welded joints. During battle, however, there is a risk of numerous projectile impacts on the vehicles. "Friction Stir Welding (FSW)" of "AA5083 aluminum-alloy" has become increasingly popular in the recent years due to its superior "mechanical characteristics", and "weldability". Compared to conventional "fusion welding processes", FSW offers numerous merits, including "high weld quality", "remarkable mechanical characteristics", and the "ability to weld-dissimilar materials". The use of FSW for the manufacture of "armor-grade targets" has been escalating due to its "superior weld quality", and "mechanical characteristics". AA5083 has been widely used in the manufacture of "armor-grade targets "owing to its "superior corrosion resistance", and "weldability". The use of FSW for the manufacture of "armor-grade targets" has also been found to be advantageous, as it provides "superior weld quality", and "mechanical characteristics". Additionally, FSW allows for the "welding of dissimilar materials", which is beneficial for the manufacture of "armor-grade targets". The use of FSW for the manufacture of "armor-grade targets" has also been found to lead to enhanced "survivability" when exposed to "multiple projectile impact". This is because FSW has been found to result in "enhanced weld quality", which leads to increased "strength" in the "welded joint". Additionally, "FSW" can "weld dissimilar materials", which can be beneficial in "armor-grade targets", as different materials can be used to achieve different "levels of protection". The novel-contribution of FSW for the manufacture of "armor-grade targets" is the ability to conduct "multiple projectile impact testing", which can be used to analyze the performance of the "target material". "Multiple projectile impact testing" involves the use of "multiple projectiles" of "distinct sizes", and "materials" to evaluate the "performance of the target-material". This type of testing is beneficial for "armor-grade targets", as it can provide insights into the "target's performance" under a variety of "extreme conditions". Overall, the use of FSW for the manufacture of "armor-grade targets" provides myriads of benefits, including "ameliorated weld quality", escalated in the "strength in the welded joint", and the ability to "weld dissimilar materials". Additionally, FSW can be used to conduct "multiple projectile impact-testing", which can provide valuable insights into the "performance of the target material". Thus, the application of "FSW" for the manufacture of "armor-grade targets" can be beneficial for enhancing the "survivability" of the "target-material". There was not much research on the "ballistic behaviour" of "thick welded targets". By studying the above-mentioned possibilities, the authors have strived to find the effect of welding on ballistic behaviour of AA5083 targets without compromising real time projectile impact. From the summary of the previous reported work, the following interpretations were

made. There is no open literature available on the ballistic behaviour of thick FSW AA5083 targets.

i. AA5083 ballistic behaviour is largely determined experimentally with the target plate in a static state.

ii. There hasn't been any significant research on the multiple hits of AA5083 FSW joints.

iii. The impact of "residual stress" on the ballistic behaviour of FSW targets has a lot of potential especially numerical simulation.

### 1.1 Objectives of the currents study

i. To prepare the defect free AA5083 FSW joints for conducting the experimental ballistic.

ii. The "numerical", and "experimental investigations" into the effects of single and multiple strikes on the "ballistic behaviour" of AA5083 targets. The effect of residual stress on ballistic behavior will be studied.

iii. Both numerical and experimental techniques were used to investigate the failure mechanisms and residual velocity of the targets.

iv. For the same target thickness, the numerical results for "BM", and "FSW joints" are compared to the experiment results for the same target thickness.

## 2. Preparation of AA5083 BM and FSW targets for experimental

AA5083 (chemical composition: Mg 4.20%, Si 0.40%, Cu 0.10%, Zinc 0.10%, Cr 0.22% and remaining Al) was processed by rolling process. Prior to FSW, the AA5083 plates were chopped to 120×60×6 mm thick. To make square FSW joints, a High Speed Steel FSW tool with a cylindrical pin profile is needed. The FSW tool has a pin diameter (5.3 mm), a shoulder diameter (17.7 mm) and pin length (5.6 mm). To obtain defect-free weld connections, a tool rotation speed of 680 rpm and a welding speed of 146 mm/min were used [26]. The defect-free AA5083 FSW joints are shown in Fig 1.

### 2.1 Experimental ballistic test on AA5083 targets

Fig 2 shows the representation of the ballistic experiment setup, and the ballistic test was conducted with a 7.62 mm hard steel core projectile. The projectile velocity is varied by changing the total of gun-powder in the cartridge.

AA5083 BM and FSW joints with dimension $120 \times 120 \times 6$ mm are taken for the ballistic experiments. MIL-DTL-32333 specified the initial projectile's velocity for AA5083 plates. By determining the projectile and target masses prior to and following the ballistic test, the "residual velocity" of the targets may be calculated. An electronic weighing machine was used to measure the target's initial and final weight with an accuracy of 0.005 g. At a starting velocity of 450 m/s, BM and FSW target plates are subjected to ballistics tests. Eq 5 illustrates how the Recht-Ipson analytical model is used to calculate the "projectile's residual velocity". The $M_p$ and $M_e$ represent the target masses prior to and following impact, while $V_f$ and $V_i$ represent the projectile's initial and residual velocities, respectively.

$$V_f = \frac{M_p}{M_{P+M_e}} V_i \tag{1}$$

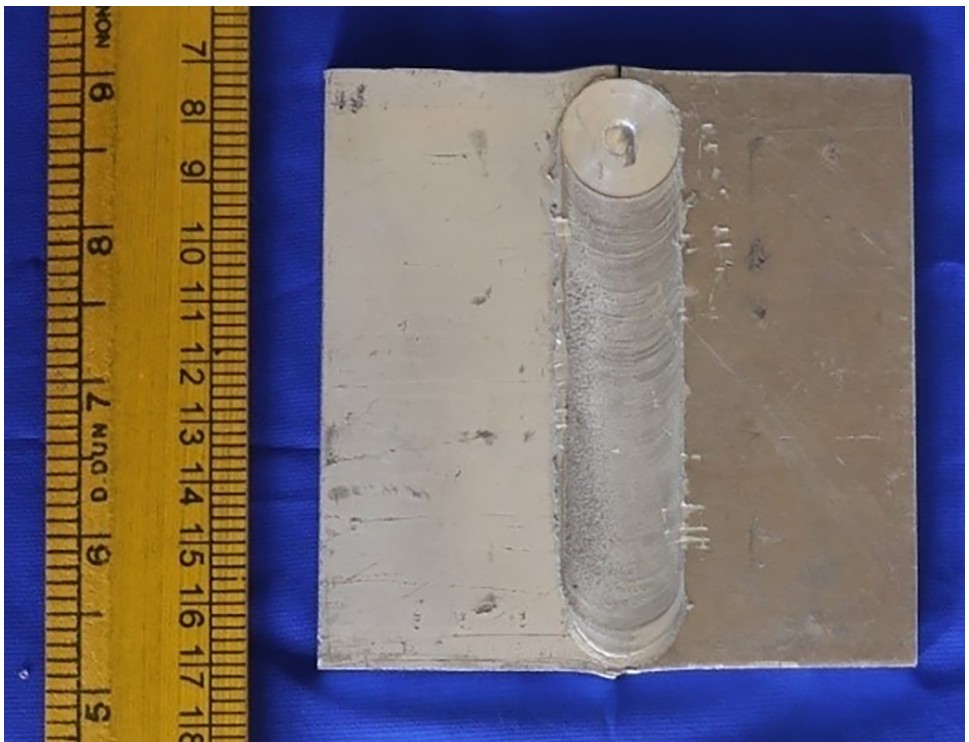

**Fig 1. Defect free AA5083 FSW joints.**

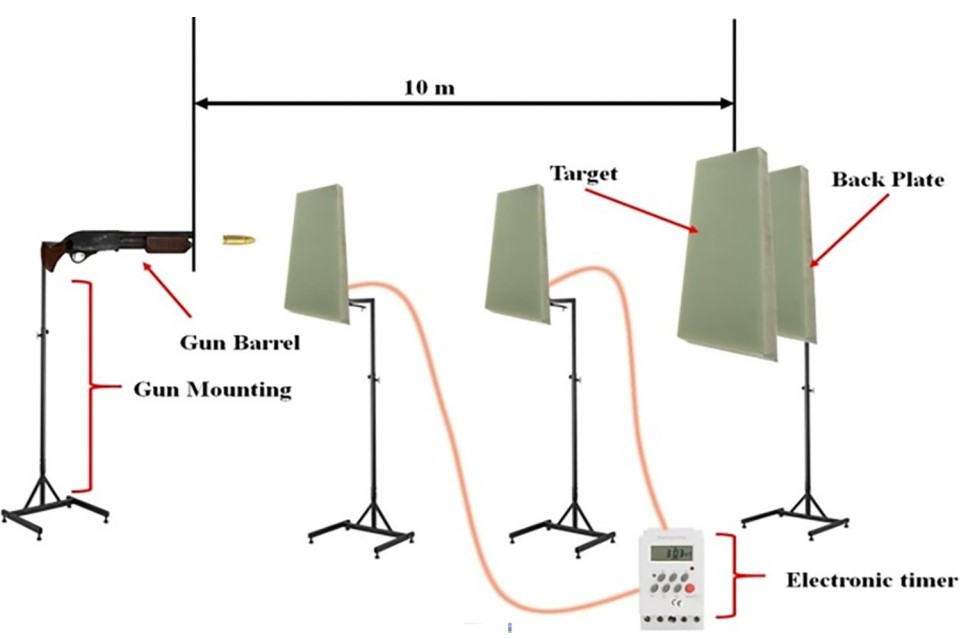

**Fig 2. Ballistic experimental setup.**

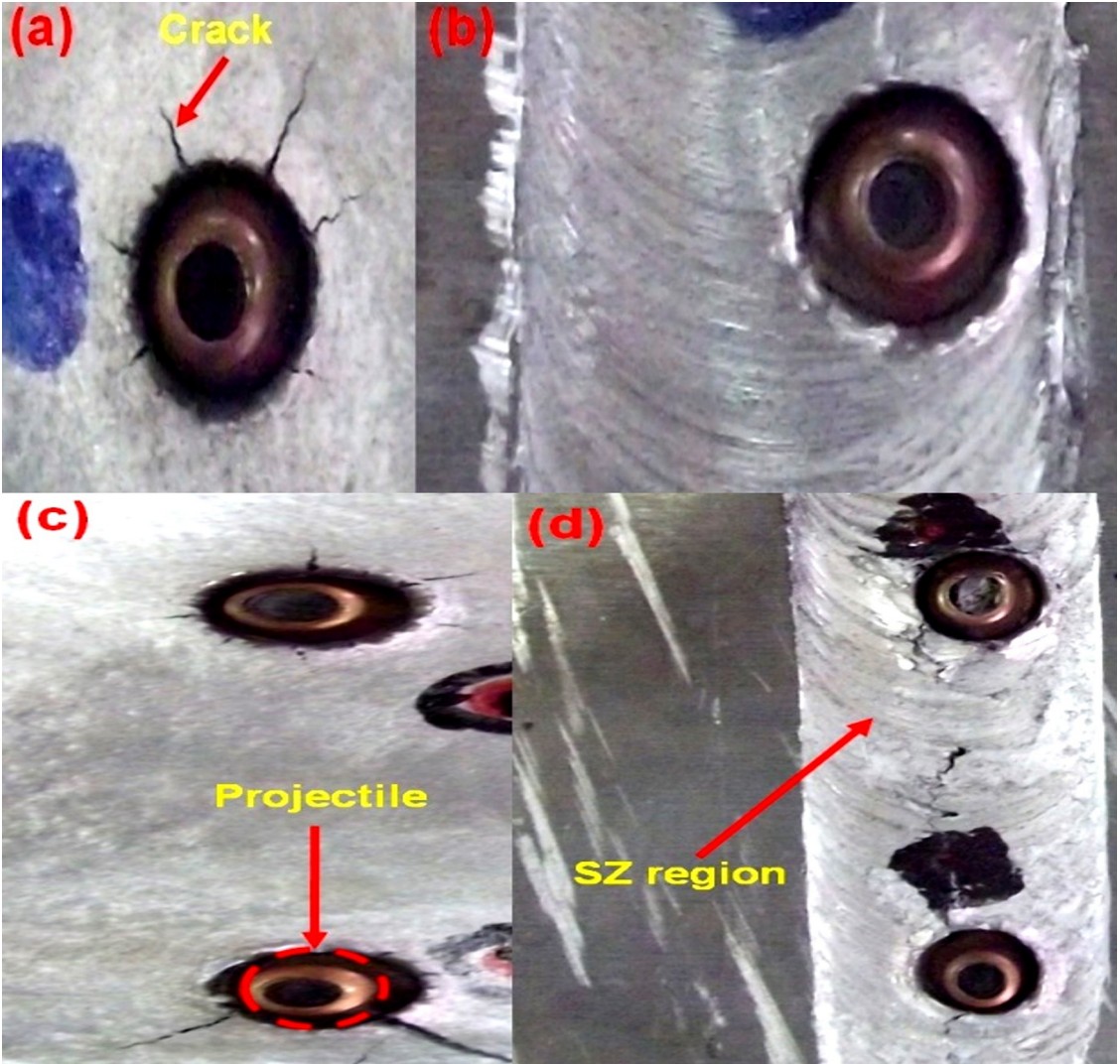

**Fig 3.** Ballistic tested AA5083 targets (a) BM-single hit (b) BM-multiple hit (c) FSW-single hit (d) FSW-multiple hit.

### 2.2 Ballistic behaviour of BM & FSW

The ballistic tested targets of AA5083 targets are shown in Fig 3(A)–3(D). On the front face of "BM targets", ductile hole failure with cracks were found. The impact impulse of the projectile overcomes the dynamic shear strength of the target material was the major reason for this type failure. Fig 3(C) and 3(D) shows the ballistic tested FSW targets. As like BM, the FSW target also failed by ductile hole failure with cracks along the SZ region.

### 2.3 Numerical analysis

ABAQUS is a "finite element code" that is used to run the "numerical simulations". The 3-D structural FE model of AA5083 target's (100 x 100 mm) has a thickness of 5 mm was used for analysis. The target was modelled as a deformable object, whereas the projectile was modelled as a rigid object. The projectile and target have a predetermined kinematic contact algorithm. The projectile's exterior surface makes contact with the crater of the target during penetration.

As a result, the projectile's outer surface is modelled as master surface, while the targets were modelled as a slave. Due to the small target thickness, it was thought that there was little to no friction between the bullet and the target. Eight node brick element (C3D8R) is used for creating FEA model.

**2.3.1. Material models.** The mechanical behaviour and the mechanics involved during the impact loading of a material involves complex phenomenon like strain hardening, plastic flow, thermal softening and fracture. The Johnson-Cook Model is a thermoviscoplastic model to model the behavior of the material over a wide range of temperatures and strain rates [24].

Johnson-Cook mode is given as;

$$\bar{\sigma} = \left[A + B(\bar{\varepsilon}^{pl})^n\right]\left[1 + C\ln\left(\frac{\bar{\varepsilon}^{pl}}{\varepsilon_0}\right)\right]\left(1 - \hat{T}^m\right) \tag{2}$$

Where $A$, $B$, $n$, $C$ and m are material factors; $\bar{\varepsilon}^{pl}$ is the equivalent plastic strain; $\varepsilon_0$ is the reference strain rate of the material, and $\hat{T}$ is the non-dimensional temperature given as;

$$\hat{T} = (T - T_0)/(T_{melt} - T_0) \tag{3}$$

$T_0 \leq T \leq T_{melt}$ Where $T$ is the current temperature, $T_{melt}$ is the melting point, and $T_0$ is the ambient temperature.

The failure of the target occurs when the damage parameter, D reaches unity.

$$D = \Sigma\frac{\Delta\bar{\varepsilon}^p}{\bar{\varepsilon}_f^p} \tag{4}$$

where, $\Delta\bar{\varepsilon}^p$ is the incremental equivalent plastic strain that occur during the "integration cycle", and $\bar{\varepsilon}_f^p$ is the "critical failure strain".

Johnson-Cook's model accounts for the implication of stress tri-axiality, strain rate effect, and the effect of temperature on the "equivalent strain to fracture". Equivalent strain to fracture is given by,

$$\bar{\varepsilon}_f^{pl} = \left[D_1 + D_2\exp\left(D_3\frac{\sigma_m}{\bar{\sigma}}\right)\right]\left[1 + D_4\ln\left(\frac{\bar{\varepsilon}^{pl}}{\varepsilon_0}\right)\right]\left(1 + D_5\hat{T}\right) \tag{5}$$

Where $D_1 - D_5$ are material parameters, $\sigma_m/\bar{\sigma}$ is the stress triaxiality ratio, and $\sigma_m$ is the mean stress. A, B, and n can be determined using uni-axial stress test, while D1, D2, and D3 were determined using uni-axial tension tests on notched specimens [27]. While m and D5 were obtained using a uni-axial tension test at extremely high temperatures, the material parameters C and D4 were found using Hopkinson pressure bar tension testing [27]. The material parameters and Johnson Cook dynamic failure model used in the present study are shown Tables 1 and 2.

**2.3.2 Meshing.** To begin, mesh convergence research was conducted in order to establish the best mesh size for the current situation. The model for the aluminum plate was built using around 9345 C3D8R hexahedral elements, whereas the model for the bullet was built with 2207 tetrahedral elements. Mesh convergence studies are used to create a fine mesh of size 0.005 in the region of the penetration zone. The model of the plate was built using 5490 C3D8T elements for similar and different welded plates. The mesh model of the welded plates and the projectile are shown in Fig 4(A) and 4(B).

Between the plate and the projectile tip, a contact interaction was defined. To apply fine mesh around the interface zone, the contact region was partitioned from the base plate. The mesh size plays a crucial role in generating the majority of the result. The element's size was

**Table 1. Material parameters for the Johnson-Cook model [27,28].**

| Description | AA5083 | Steel 4340 |
|---|---|---|
| Young's modulus(E) | $72\times10^3$MPa | $205\times10^3$MPa |
| Poisson's ratio (*V*) | 0.3 | 0.3 |
| Density(*ρ*) | 2660 kg/m$^3$ | 7830 kg/m$^3$ |
| Yield stress (A) | 167 N/mm$^2$ | 1430 N/mm$^2$ |
| Strain-hardening constant(B) | 596 N/mm$^2$ | 2545 N/mm$^2$ |
| Viscous effect(n) | 0.55<br>0.001 | 0.7<br>0.01 |
| Thermal-softening constant (m) | 1 | 1 |
| Melting temperature ($T_{melt}$) | 620°C | 1520°C |
| Transition temperature ($T_0$) | 20°C | 20.2°C |

minimized and reduced to achieve more precise and accurate results. The mesh size of AA5083 target plate in the weld region is 0.01 and for steel it is 0.001.

*2.3.3 Finite element boundary conditions.* The plate was analyzed to ensure a close correlation with the samples used in the experiments. Therefore, the outer edges of the target plate are considered as a fixed end, similar to that of the experiments that are in a static position. While projectiles are being dynamically moved by applied velocity in order to observe impact damage. The base must be fixed rigidly without movement on any side, with all four edges completely constrained in all the DOF. As the projectile hits the target's surface, an initial velocity of 450 m/s is applied. In case of FSW plate, the surface heat flux was calculated as per FSW process parameter [29]. The heat input in the welded region was 322W [26] and it was given to all over the weld region. The velocity for multiple hits was set at 450 m/s, as determined by literature [30,31].

# 3. Results and discussions

## 3.1 Numerical ballistic performance of BM and welded targets

Ballistic impact testing is performed on the base material AA5083 targets under conditions where both a single and multiple projectiles are fired. Performance is expressed in terms of the targets' kinetic energy absorbed upon collision at 450 m/s. It immediately includes the projectile's velocity being transferred to the target's residual velocity. Fig 5 shows the two kinetic energy states that represent the effects of a single impact and multiple hits. In both circumstances, it is simulated as hitting the target in the middle.

The effects of the steel projectile on the single hit AA5083 BM are shown in Fig 6(A) and 6 (B).The target completely failed by ductile failure with additional plastic deformation [31–33]. Fig 7(A)–7(C) depicts the several impacts of the steel projectile on the AA5083 BM target.

The multiple hits of the projectile on the AA5083 target failed by ductile failure, similar to a single hit. The projectile's effect is extended to investigate the target against several hits. The effect was demonstrated by bending the target plate over several colour schemes. The crack grows gradually due to the elliptical hole [32–34]. By causing ductile fracture formation, the

**Table 2. Parameters for "Johnson-Cook damage model"[27].**

| Material | $D_1$ | $D_2$ | $D_3$ | $D_4$ | $D_5$ |
|---|---|---|---|---|---|
| Aluminium 5083 | 0.026 | 0.26 | -0.34 | 0.14 | 16.8 |
| Steel 4340 | 0.05 | 3.4 | 2.1 | 0.002 | 0.61 |

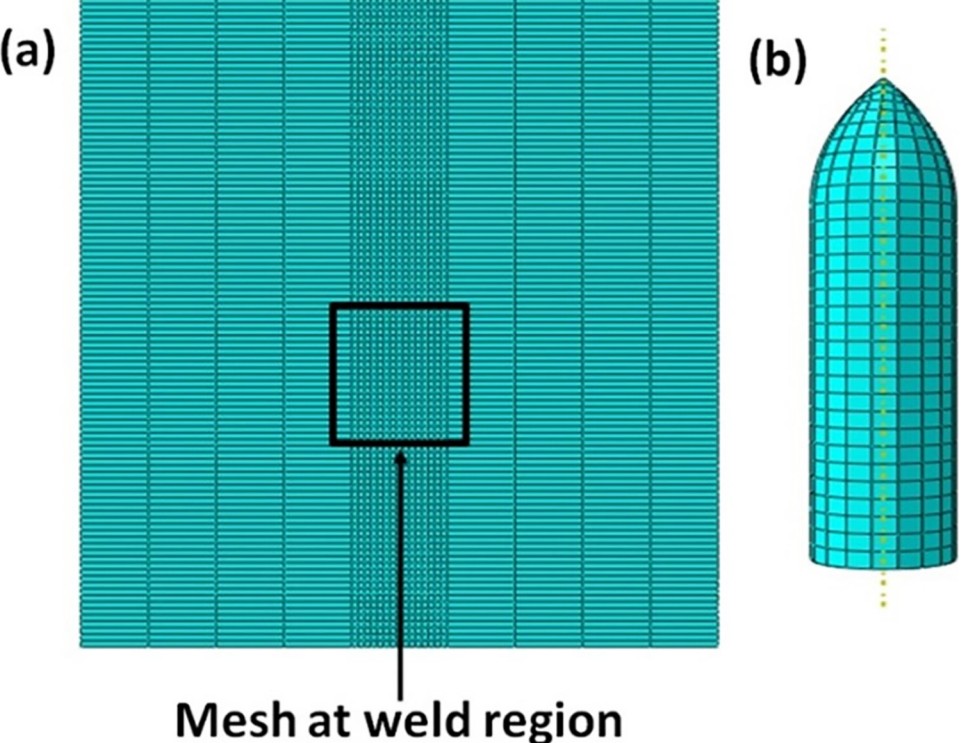

**Fig 4.** Mesh model (a) AA5083 FSW plate (b) projectile.

residual velocity produced was able to extend over the material and break it up. By contacting the second projectile with the same speed as the first, the process was repeated. According to the graph, the initial 450 m/s had an excellent decrease in kinetic energy; however, the second impact of velocity needed less time to nullify the kinetic energy. The target was able to withstand the effects of the second hit as a result. This indicates that in the early impact zones, the material is strong enough to resist the entering projectile.

### 3.2 FSW AA5083 targets

AA5083 material property is applied to entire target to evaluate the ballistic performance. However, a "meshing size" is assigned from the "base target" with "applied heat flux", and the welded region is given the welded material model. To produce reliable readings similar to the experimental method, heat flux in the welded zone is required [34–36]. During the weld metal, this same heat flow is modelled to the real heated area by refining the grains as the material melts. As a result, the hardness in that area naturally decreases. Fig 8(A) and 8(B) expose the failure regions of single hit target.

The projectile forms the more ductile hole failure in the welded region. It is due to the effect of residual stress formed due to the welding. The projectile hole formation on the front side is completely different from the BM. Fig 9(A)–9(C) depicts the multiple hits of the steel projectile on the AA5083 FSW targets. The elliptical hole creation is larger than the BM.

The projectile's several hits on the FSW AA5083 target failed due to ductile failure and significant plastic deformation. However, the large separation of the material is observed in the between the two projectile as shown Fig 9. The second projectile easliy penetrated in to the traget as compared to the first projecilte. The initial hole formation is the major reason for

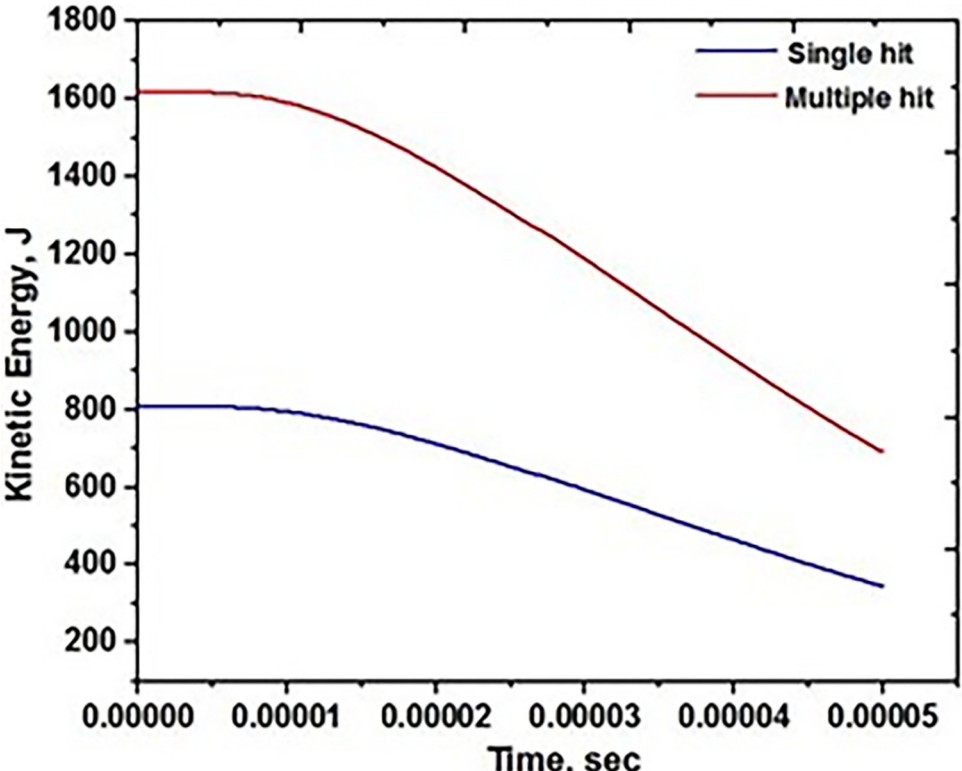

**Fig 5. Ballistic performance on base metal.**

formation of large gap between the two holes. The kinetic energy in the target's surface is determined using the data provided in Fig 10, which also measures the value with several strikes on the surface target.

As the velocity of the projectile steadily drops, bending in the shape of the target is seen in the single impact region. The material was to achieve ductile failure by widening the impact zone. As a result, a single bullet generates ductile development with limited penetration,

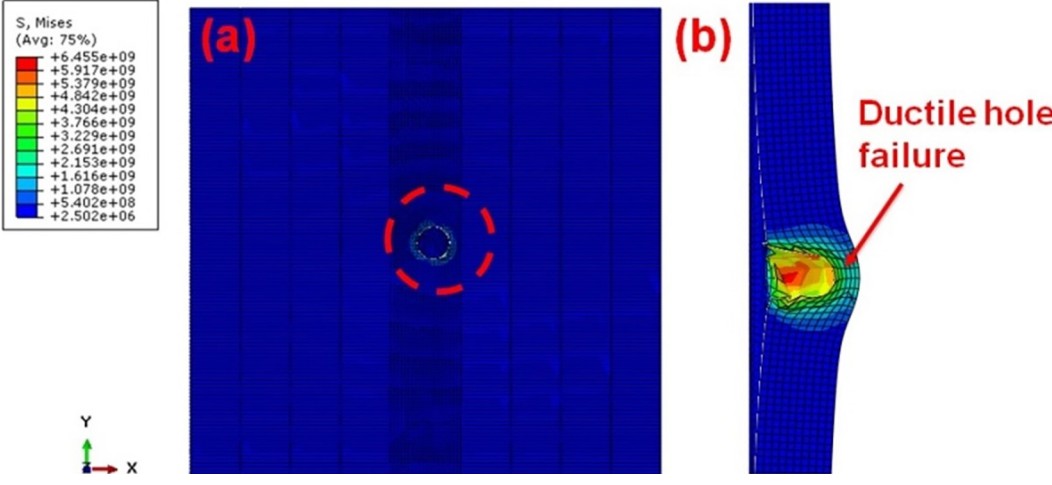

**Fig 6.** BM Single hit view (a) front (b) sectioned.

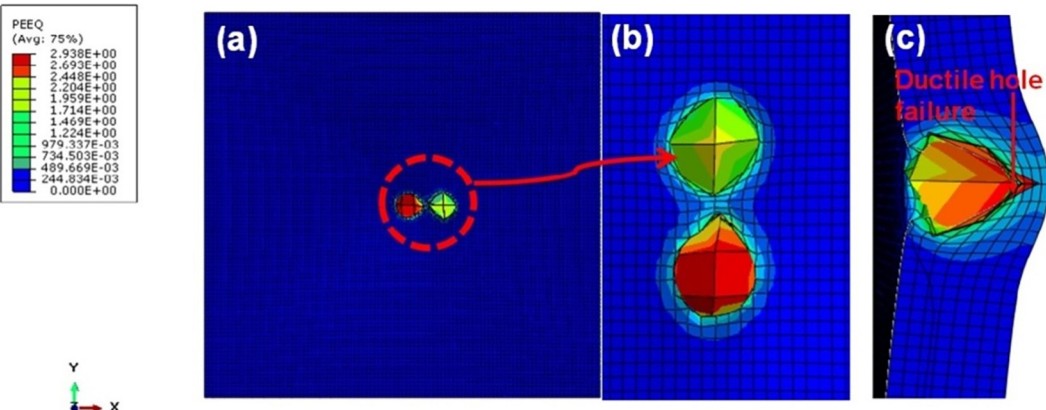

**Fig 7.** BM multiple hit (a) front view (b) zoomed front view (c) sectioned view.

expressing the hardness of the target with such velocity [32–34]. As the impact of the target rises across the region, so does its residual velocity or absorbing energy in Joules on the target's surface. This can be shown in the case of repeated impact testing. The development of absorbing energy in the target is caused by the approach of several bullets. The absorbing energy increases with each increase in projectile impact. The absorption energy rose faster over the welded region than the original hit. It is caused by fracture creation and a lack of energy flow to the entire surface, resulting in a crack over the area. As similar impacts occur in the nearby locations, the material's resistance decreases [35–36]. The second one shows a progressive slope on the target's kinetic energy while it stays still for a short time and then slowly goes down. It's because the target's grain refining has reduced its hardness, making it less resistant to future impacts in that location.

Experimental and numerical results are compared in Fig 11. In both experimental ($V_r$ = 179.2 m/s) and numerical ($V_r$ = 196.66 m/s) results, the average residual velocity for the BM was very close to that of the BM. The less variation could be because of the uncertainty that comes up during experiments and the assumptions that are made for the numerical simulation. Related to the measurements observed for BM, the FSW target showed the same residual velocity coincidence.

The measured residual velocity ($V_r$ = 283.15m/s) is 5.2% lower than the numerically calculated residual velocity ($V_r$ = 268.2m/s). Higher plastic deformation along with thermal

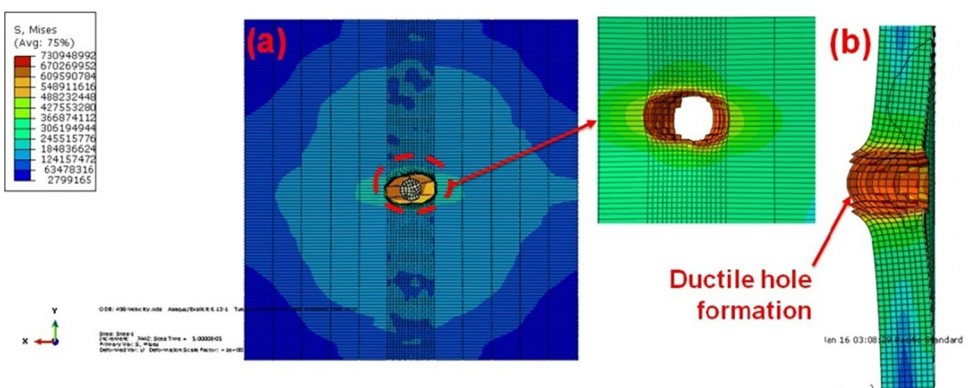

**Fig 8.** FSW Single hit (a) front view (b) sectioned view.

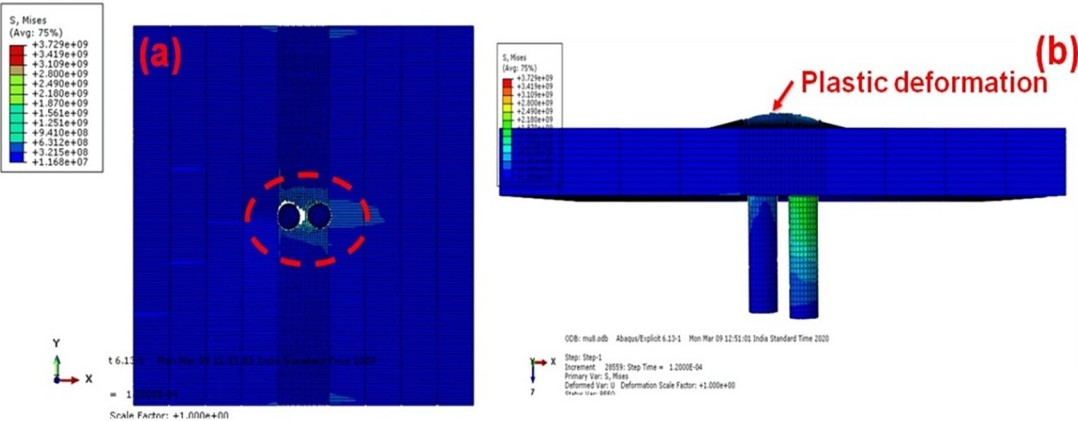

**Fig 9.** FSW Multiple hit (a) front view (b) sectioned view.

softening effects dueto FSW process caused the target to lose its ballistic resistance. The numerical technique's prediction of residual velocities in both scenarios (BM and FSW) demonstrated good agreement with the experimental method. The average residual velocity decreased from 283.155 to 196.2 m/s (FSW) (BM). Consequently, the BM target has stronger ballistic resistance than the FSW target.

From the comparable studies, it has been unveiled that the "experimental", and "numerical studies" of "tungsten-rod" impact on "Alumina/Aluminium 603 armour steel "have likewise

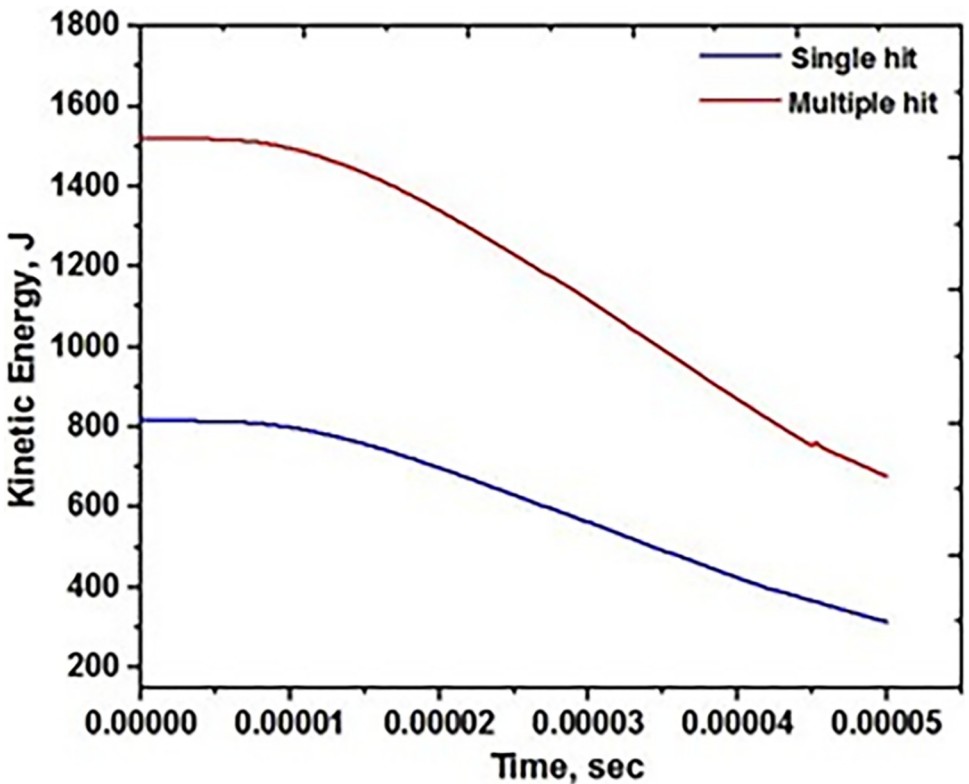

**Fig 10. Ballistic performance on FSW.**

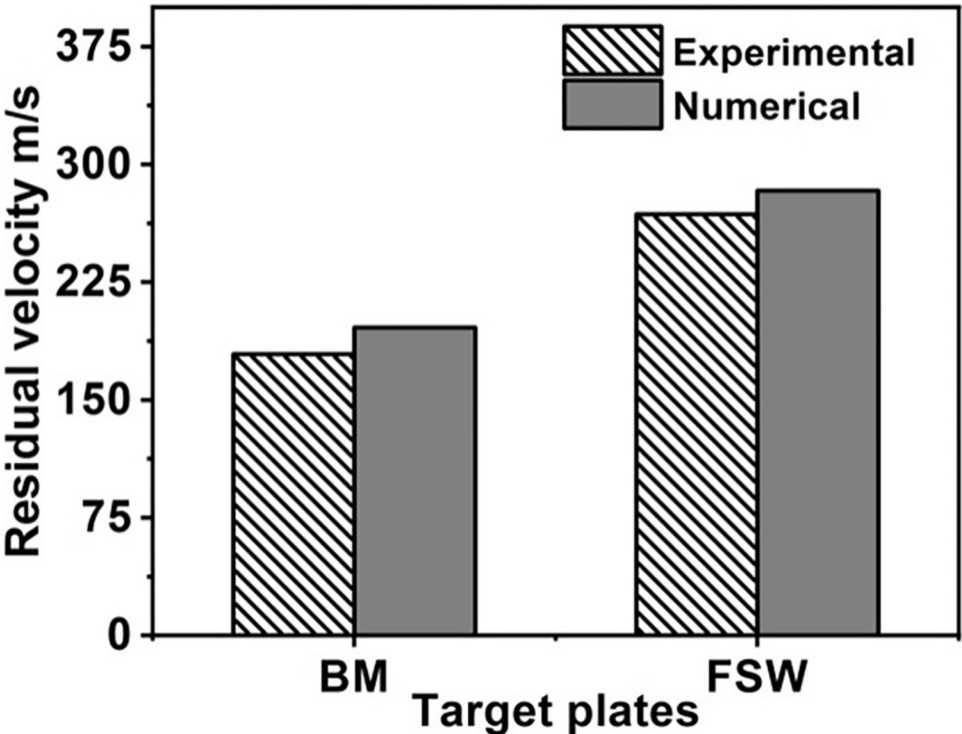

**Fig 11. Experimental and numerical results.**

been conducted by Jinzhu et al. [37]. Furthermore, it was reported that there was a "linear relationship" between "depth of penetration", (DOP) and "ceramic thickness" between the "steel back plate", and "ceramic thickness". Increases in "ceramic thickness" resulted in a higher mass and differential efficiency factor. A study by Madhu et al. [38] examined the "ballistic performance" of 95% and 99.5% "alumina tiles" against a 12.7 mm AP projectile. As a function of the "ceramic thickness", and the "velocity of the projectile", a "ballistic efficiency factor (BEF)" was calculated based on Frank's work [39]. There was a higher ballistic performance for alumina 99.5% as compared to alumina 95%. With increasing thickness, alumina 99.5%'s ballistic efficiency factor decreased and alumina 95%'s BEF increased. By impacting boron carbide (B4C) tiles with a 7.62mm AP projectile, Savio et al. [40] reported increased differential efficiency factor (DEF) with velocity increase. An impact test was conducted by Rosenberg et al. [41] using AP projectiles of "0.3", "0.5", and "14.5 mm thickness" on "alumina tiles (AD 85)". AP projectiles of all types exhibited a "linear relationship" between "residual penetration", and "ceramic tile thickness". Various projectiles exhibited a "linear relationship" between "residual penetration", and "ceramic tile thickness" according to Rosenberg and Dekel [42], indicating that tile thickness does not affect ballistic efficiency. It has been noted that there is inconsistency in the outcomes stated in the literature due to the difficulty in reproducing "ceramic tiles" with consistent "material properties", and the effects associated with "thick", or "thin tiles". In order to determine the ballistic efficiency of tiles, it is important to choose the tile thickness carefully. Similarly, Wang et. al. [43] found that the size of the hole, the depth of the crater, and the depth of penetration were determined by the design of the shaped charge. Finite element modeling provides a means of studying high-speed impacts using numerical simulations. In most numerical studies, ceramic materials are modeled using "JH-1", and "JH-2 Johnson-Holmquist material models". Physiological material models such

as Johnson-Holmquist predict the durability and failure of ceramics. When applied to materials, the JH-1 model predicts instantaneous failure, but the JH-2 model is capable of predicting damage in materials. A "finite element method" was used by Pawar et al. [44] to compare the "ballistic performance" of "Al2O$_3$/Al5083 ceramic armours", and "A1N/Al5083 ceramic armours". JH-2 material model was used to define ceramic materials. When 7.62 mm AP bullets were impacted on the armour, AIN/Al 5083 performed better than Al$_2$O$_3$/Al 5083 in terms of ballistic performance. The "ballistic performance" of SiC/Al armour was compared numerically to Al2O3/Al armour by Venkatesan et al. [45]. For both ceramics, the "JH-2 material model" was used, and it was stated that the SiC ceramic had greater compressive strength, resulting in more damage [46–48]. It is therefore evident that SiC/Al ceramics perform better in ballistics tests than Al$_2$O$_3$/Al ceramics [49–51]. Based on JH-2 based finite element simulations, it can be seen that the material behavior under ballistic conditions could be accurately predicted using finite element simulations [52–54].

Thus, through the consideration of the above-mentioned possibilities, it's been evident that the welding affects the ballistic behavior of AA5083 targets without impairing the real-time impact of the projectile. As military vehicles have been constructed from steel plates and welded joints [55–57]. There is, however, a possibility that multiple projectiles might strike a vehicle during a battle.

Hence, the aim of this study was to investigate the ballistic resistance (BR) of AA5083 aluminum alloy targets made of base material (BM) and friction stir welded (FSW) joints. The study involved building a finite element model to predict the kinetic energy absorption and target deformation under both single and multiple projectile impact conditions. The study used 7.62mm Hard Steel Core (HSC) projectiles produced from Steel 4340, and the targets were analyzed using commercially available Abaqus Explicit software for Finite Element Analysis. Results showed that the kinetic energy generated and surface residual velocity increased as the number of projectile strikes increased [58–60]. An experimental ballistic test was conducted to validate the numerical results, and the analytical Recht-Ipson model was used to determine each target's experimental residual velocity [61–63]. It was found that weldments perform less well (30%) than BM targets due to the occurrence of plastic deformation during welding [64–66]. Both numerical and experimental techniques were used to investigate the failure mechanisms and residual velocity of the targets, and a correlation between residual velocities was found [67–69]. Furthermore, the study aimed to prepare defect-free AA5083 FSW joints for conducting the experimental ballistic and to compare the numerical and experimental results of BM and FSW targets under the same thickness [70–72]. The study found that both BM and FSW targets exhibited a non-linear variance in kinetic energy, and material fragmentation was the ultimate failure condition in the impact region for multiple projectile hits [73]. The residual velocities of BM and FSW targets were 196 m/s and 283.155 m/s, respectively, based on the experiments, with BM outperforming FSW targets in terms of ballistic resistance. The computed residual velocities were close to the experimental observations, indicating that explicit numerical coding can be utilized to anticipate target ballistic resistance, potentially saving money on expensive ballistic experiments [74].

In the comparison of the outcomes of our investigation with prior findings in the existing literature, we confirmed that the conclusions were comparable with those of several existing works. For instance, Dannemann et al. (2016) investigated the deformation response of fusion and friction stir welded aluminum plates using digital image correlation (DIC) [75]. The results showed that the deformation patterns of the two welding techniques differed significantly. Fusion welding resulted in more localized deformation while friction stir welding produced a more uniform deformation pattern. DIC was found to be a valuable tool for comparing the deformation response of different welding techniques. Praveen et al. (2022)

conducted numerical and experimental investigations to study the ballistic behavior of AA7075 thick plates under different target thicknesses and solution treatments [76]. The results showed that the solution treatment improved the BR of the plates, and thicker plates exhibited better ballistic performance. The numerical simulations agreed in context with the findings from the experiment, signifying the accuracy of the numerical model in predicting the ballistic performance of the plates. Magarajan and Kumar (2023), friction stir processing of aluminum (AA-6061/B$_4$C) surface composite has promising ballistic characteristics [77]. The results showed that the composite had a remarkable ballistic performance in comparison with the AA-6061 due to the homogenization of the microstructure and the dispersion of B$_4$C particles. The study also revealed that the processing parameters had a significant impact on the microstructure and ballistic performance of the composite. Rahman et al. (2018) examined the implications of layering configuration on the high-speed impact behavior of laminated aluminum/steel panels [78]. The results showed that the panel with alternating layers of aluminum and steel exhibited the best ballistic performance, as it absorbed more energy and deformed uniformly compared to other layering configurations. The numerical simulations agreed well with the outcomes of experiment, illustrating the accuracy of the numerical model in analysing the behavior of laminated panels under high-speed impact.

## 4. Conclusions

The numerical and experimental behaviour of "BM", and "FSW", "AA5083 targets" against single and multiple projectile hits was determined.

a.  The target's surface kinetic energy increases as the number of projectile hits increases. Both BM and FSW targets exhibited a non-linear variance in kinetic energy.

b.  Multiple projectile hits demonstrate that it takes more time to cancel the projectile's velocity. Material fragmentation was initiate to be the ultimate failure condition in the impact region. At process conditions of 1100 rpm and 50 mm/min, defect-free FSW AA5083 targets were produced

c.  The "residual velocities" of the "BM", and "FSW targets" were 196 m/s and 283.155 m/s, respectively, based on the experiments. When compared to the FSW target, the BM exhibited a 30% lower residual velocity. As a result, the BM target outperformed the FSW target in terms of ballistic resistance.

d.  The computed residual velocities of both targets were quite close to the experimental observations. As a result, explicit numerical coding can be utilised to anticipate target ballistic resistance, potentially saving money on expensive ballistic experiments.

## Acknowledgments

The authors express their gratitude to Princess Nourah bint Abdulrahman University Researchers Supporting Project number (PNURSP2023R61), Princess Nourah bint Abdulrahman University, Riyadh, Saudi Arabia.

## Author Contributions

**Conceptualization:** S. Balaji, S. Dharani Kumar, U. Magarajan, S. RameshBabu, S. Ganeshkumar, Shubham Sharma.

**Formal analysis:** S. Balaji, S. Dharani Kumar, U. Magarajan, S. RameshBabu, S. Ganeshkumar, Shubham Sharma, Shaimaa A. M. Abdelmohsen.

**Funding acquisition:** Shubham Sharma, Shaimaa A. M. Abdelmohsen, Sayed M. Eldin.

**Investigation:** S. Balaji, S. Dharani Kumar, U. Magarajan, S. RameshBabu, S. Ganeshkumar, Shubham Sharma.

**Methodology:** S. Balaji, S. Dharani Kumar, U. Magarajan, S. RameshBabu, S. Ganeshkumar, Shubham Sharma.

**Project administration:** Shubham Sharma, Sayed M. Eldin.

**Supervision:** Shubham Sharma, Indranil Saha, Sayed M. Eldin.

**Validation:** Shaimaa A. M. Abdelmohsen.

**Writing – original draft:** S. Balaji, S. Dharani Kumar, U. Magarajan, S. RameshBabu, S. Ganeshkumar, Shubham Sharma.

**Writing – review & editing:** Shubham Sharma, Shaimaa A. M. Abdelmohsen, Indranil Saha, Sayed M. Eldin.

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
