## [Decision Letter · Decision Letter 0]

26 Jan 2023

PONE-D-22-32018Experimental and numerical investigation of an AA5083 targets against multiple projectile impact: A comparative analysisPLOS ONE

Dear Dr. Eldin,

Thank you for submitting your manuscript to PLOS ONE. After careful consideration, we feel that it has merit but does not fully meet PLOS ONE’s publication criteria as it currently stands. Therefore, we invite you to submit a revised version of the manuscript that addresses the points raised during the review process.

Please, address all the comments made by the reviewers.

We look forward to receiving your revised manuscript.

Kind regards,

Antonio Riveiro Rodríguez, PhD

Academic Editor

PLOS ONE

Journal Requirements:

Reviewers' comments:

Reviewer's Responses to Questions

**Comments to the Author**

1. Is the manuscript technically sound, and do the data support the conclusions?

Reviewer #1: Partly

Reviewer #2: Yes

Reviewer #3: Partly

2. Has the statistical analysis been performed appropriately and rigorously? 

Reviewer #1: N/A

Reviewer #2: Yes

Reviewer #3: Yes

3. Have the authors made all data underlying the findings in their manuscript fully available?

Reviewer #1: Yes

Reviewer #2: Yes

Reviewer #3: Yes

4. Is the manuscript presented in an intelligible fashion and written in standard English?

Reviewer #1: Yes

Reviewer #2: Yes

Reviewer #3: Yes

5. Review Comments to the Author

Reviewer #1: The authors perform experimental and numerical work on AA5083 against ballistic impact. The work should be improved for publication.

- The description of the behavior of AA5083 is quite imprecise. The JC Model is a thermoviscoplastic model to model the behavior of the material over a wide range of temperatures and strain rates. It should be rewritten. The use of "" is not understood in very clear terms.

- Figure 2 is dispensable. In addition, the following sentence is included in the parenthesis: "The heat input in the welded region was 322W [27] and it was given to all over the weld region". This has been modeled?

- Have they used a damage initiation and evolution model in ABAQUS or was it a Shear Failure?

- The mesh in Figure 6 looks overly extorted.

- The concepts of damage mechanisms are wrong.

- The paper is quite confusing, it should start with the experimental part.

In the present state it is not advised to be published.

Reviewer #2: In my oprinion, no new finding can be found in the present study. Therefore, It should be rejected. The other resaon is that the content of te present study is not well fitted wit the aim and scope of the Journal.

Reviewer #3: 1. In the Title FSW title should be included

2. Replace the term Base metal with Base material

3. “ It was determined that weldments perform less well (30%) as compared to BM targets” reason can be included in the abstract

4. Scale bar missing in the Fig.3

5. What are the process parameters considered for the current study

6. Many typo errors are there in manuscript please check.

7. For armoured vehicles require thicker section base material. But current study was lesser thickness

8. Reframe article to publication format it looks like a report

6. PLOS authors have the option to publish the peer review history of their article (what does this mean?). If published, this will include your full peer review and any attached files.

Reviewer #1: No

Reviewer #2: No

Reviewer #3: No

---

## [Author Response · Author response to Decision Letter 0]

14 Feb 2023

07.02.2023

Dear Prof. (Dr.) Editor-in-chief,

Thank you for considering the manuscript entitled, “Experimental and numerical investigation of an AA5083 targets against multiple projectile impact: A comparative analysis”, for the publication in PLOS ONE. I am grateful to you and the reviewers for the valuable suggestions provided. I like to resubmit our revised version of the manuscript by adding response to all your comments. Below please find the answers and actions taken to address these comments. All the suggestions are incorporated and highlighted with the BLUE COLOR in manuscript. 

NOTE: All the necessary changes/added sentence has been shown in the BLUE COLOR.

The locations of these changes have been mentioned, where possible, in the action points that respond to each reviewers’ comments. Here are the responses to the reviewer comments:

AUTHOR RESPONSE TO REVIEWER AND EDITOR COMMENTS

Manuscript Number: PONE-D-22-32018

Article title: Experimental and numerical investigation of an AA5083 targets against multiple projectile impact: A comparative analysis

Journal: PLOS ONE

The manuscript has been thoroughly modified and improved the quality of the content to meet the standards of the Journal. All the suggestions made by the learned referees are included in the revised manuscript. We are extremely thankful to the referees & editor(s) for their constructive comments and appreciation. 

Response to Reviewer’s Comments

The authors are grateful to the reviewers for their suggestions that have all contributed to improving the manuscript. Once again, the authors are extremely thankful for the observations and the comments of the reviewers. All the comments are appropriately addressed and now the quality of the article has been appreciably enhanced before the consideration for publications. The rebuttal file is enclosed indicating the revisions incorporated in the article as suggested. The revisions are carried out in BLUE COLOR in the text of the manuscript for better visibility to the reviewers and as well as to the editor. We have made the modifications as per their suggestions in the revised manuscript and changes are also marked up using the “BLUE COLOR” function.

All in all, the authors should thank the reviewers for their meticulous observations in reviewing the article. All the issues raised by the authors are appropriately addressed as stated in the following table,

Author Response to Reviewer’s Queries

We express our sincere thanks to the reviewers and editor for their valuable suggestions to improve our manuscript. The following are the detailed author’s response (AR) for reviewer’s queries (RQ).

Reviewer #1: Queries and authors response

The authors perform experimental and numerical work on AA5083 against ballistic impact. The work should be improved for publication.

RQ:1 The description of the behavior of AA5083 is quite imprecise. The JC Model is a thermoviscoplastic model to model the behavior of the material over a wide range of temperatures and strain rates. It should be rewritten. The use of "" is not understood in very clear terms.

AR:1 As per reviewer comment, the above statement has been rewritten. 

RQ:2 Figure 2 is dispensable. In addition, the following sentence is included in the parenthesis: "The heat input in the welded region was 322W [27] and it was given to all over the weld region". This has been modeled?

AR:2 As per reviewer comment, Figure 2 has been removed and heat input was taken from the reference [27]. During simulation heat input value is assigned in stir zone region. 

RQ:3 Have they used a damage initiation and evolution model in ABAQUS or was it a Shear Failure?

AR:3 We have used Johnson-Cook material and damage model in ABAQUS.

RQ:4 The mesh in Figure 6 looks overly extorted.

AR:4 Based on mesh convergence study, we have arrived the mesh size. Hence it was extorted.

RQ:5 The concepts of damage mechanisms are wrong.

AR:5 The concepts of damage mechanisms have thoroughly been explained based on common type of ballistic failure mechanisms.

RQ:6 The paper is quite confusing, it should start with the experimental part

AR:6 As per reviewer comment, the order has been changed. The authors have tried to refine the overall content, structure, formatting, and relevancy of the revised article, specifically in the Introductions section as per the learnt recommendations. The authors have now cited a number of relevant recent articles on related works, thereby enhancing the content's appeal and retainability among readers of scientific journals. In the revised article, the authors have attempted to refine the overall content, structure, scientific contributions, novelty, and relevance as per the learned recommendations.

Additionally, the authors have removed any unnecessary or undesirable information in the article and polished the manuscript in a more accurate and precise way, removing any junk content, and organizing it in an orderly and systematic format related to the literature studies, research gaps, problem formulation, objectives, and research methodology. By systematically supporting the present results with previous literature reviews, the outlines of the discussions have been extensively expanded and furthermore enumerated in a more comprehensive and extensive way.

All in all, the authors have excavated or removed the unnecessary matter or data from the article and thus, the manuscript has been refined, prepare and contemplate it in more precise or accurate, and thus, organized in the systematic orderly manner.

Reviewer #2: Queries and authors response

RQ:1 In my opinion, no new finding can be found in the present study. Therefore, It should be rejected. The other reason is that the content of to present study is not well fitted with the aim and scope of the Journal.

AR:1 The authors are enormously thankful to the learnt referee for their knowledgeable insights on the same. The novelty statement and scientific contributions have extensively been enumerated in revised manuscript as exhibited in the blue colour.

The authors have tried to refine the overall content, structure, formatting, and relevancy of the revised article, specifically in the Introductions section as per the learnt recommendations. The authors have now cited a number of relevant recent articles on related works, thereby enhancing the content's appeal and retainability among readers of scientific journals. In the revised article, the authors have attempted to refine the overall content, structure, scientific contributions, novelty, and relevance as per the learned recommendations.

Additionally, the authors have removed any unnecessary or undesirable information in the article and polished the manuscript in a more accurate and precise way, removing any junk content, and organizing it in an orderly and systematic format related to the literature studies, research gaps, problem formulation, objectives, and research methodology. By systematically supporting the present results with previous literature reviews, the outlines of the discussions have been extensively expanded and furthermore enumerated in a more comprehensive and extensive way.

Reviewer #3: Queries and authors response

RQ:1 In the Title FSW title should be included

AR:1 As per reviewer comment, the friction stir welding has been added in the title of the paper. 

RQ:2 Replace the term Base metal with Base material

AR:2 As per reviewer comment, the base metal has been replaced with base material. 

RQ:3 “It was determined that weldments perform less well (30%) as compared to BM targets” reason can be included in the abstract

AR:3 As per reviewer comment, the reason for less ballistic performance of weldments has been included in the abstract 

RQ:4 Scale bar missing in the Fig.3

AR:4 As per reviewer comment, the scale bar has been added in the Figure 3

RQ:5 What are the process parameters considered for the current study?

AR:5 To obtain defect-free weld connections, a tool rotation speed of 680 rpm and a welding speed of 146 mm/min were used [27].

This are the process parameter used for preparation of weldments 

RQ:6 Many typo errors are there in manuscript please check.

AR:6 As per reviewer comment, in the revised manuscript all typo errors has been corrected. As per suggestions, all the mistakes in accordance with the writing skills and usage of English language have now been polished up to a fervent extent with the assistance of the native English contributor/co-author of this article.

As per valuable suggestions received, the grammatical errors have now been removed and many sentences have been reframed so as to deliver clear interpretation in the revised manuscript.

RQ:7 For armoured vehicles require thicker section base material. But current study was lesser thickness

AR:7 Less thickness plates are used for fabrication of trailer beds in the defence tankers. Hence, our Initially study is focused on the lesser thickness plates. 

RQ:8 Reframe article to publication format it looks like a report

AR:8 As per reviewer comment, the complete articles has been reframed. Now, the manuscript has now been restructured, refined, formatted, and extensively modified or revised as per the standard template of the Journal (PLOS ONE). 

The authors have tried to refine the overall content, structure, formatting, and relevancy of the revised article, specifically in the Introductions section as per the learnt recommendations. The authors have now cited a number of relevant recent articles on related works, thereby enhancing the content's appeal and retainability among readers of scientific journals. In the revised article, the authors have attempted to refine the overall content, structure, scientific contributions, novelty, and relevance as per the learned recommendations.

Additionally, the authors have removed any unnecessary or undesirable information in the article and polished the manuscript in a more accurate and precise way, removing any junk content, and organizing it in an orderly and systematic format related to the literature studies, research gaps, problem formulation, objectives, and research methodology. By systematically supporting the present results with previous literature reviews, the outlines of the discussions have been extensively expanded and furthermore enumerated in a more comprehensive and extensive way.

All in all, the authors have excavated or removed the unnecessary matter or data from the article and thus, the manuscript has been refined, prepare and contemplate it in more precise or accurate, and thus, organized in the systematic orderly manner.

Hence, a scientific explanation of the obtained results has been refined and ameliorated up to a fervent extent. Results are enumerated, test methods are utterly described, interpretation have been corelated with results and previous literature findings. The overall summary should indicate the progress of the research and the limitations. 

Note: All the necessary changes/added sentence has been shown in the BLUE COLOR. 

Thank you very much in advance for taking your time in reviewing this manuscript. 

Sincerely, we hope you will find our revision satisfactory.

Thanks, in anticipation.

Regards,

Dr. Shubham Sharma

(Corresponding author)

---

## [Decision Letter · Decision Letter 1]

12 Mar 2023

PONE-D-22-32018R1Experimental and numerical investigation of an AA5083 friction stir welded targets against multiple projectile impact: A comparative analysisPLOS ONE

Dear Dr. Eldin,

Thank you for submitting your manuscript to PLOS ONE. After careful consideration, we feel that it has merit but does not fully meet PLOS ONE’s publication criteria as it currently stands. Therefore, we invite you to submit a revised version of the manuscript that addresses the points raised during the review process. Please, address all the comments made by the reviewers.

We look forward to receiving your revised manuscript.

Kind regards,

Antonio Riveiro Rodríguez, PhD

Academic Editor

PLOS ONE

Reviewers' comments:

Reviewer's Responses to Questions

**Comments to the Author**

1. If the authors have adequately addressed your comments raised in a previous round of review and you feel that this manuscript is now acceptable for publication, you may indicate that here to bypass the “Comments to the Author” section, enter your conflict of interest statement in the “Confidential to Editor” section, and submit your "Accept" recommendation.

Reviewer #1: All comments have been addressed

Reviewer #3: All comments have been addressed

2. Is the manuscript technically sound, and do the data support the conclusions?

Reviewer #1: Yes

Reviewer #3: Yes

3. Has the statistical analysis been performed appropriately and rigorously? 

Reviewer #1: N/A

Reviewer #3: Yes

4. Have the authors made all data underlying the findings in their manuscript fully available?

Reviewer #1: Yes

Reviewer #3: Yes

5. Is the manuscript presented in an intelligible fashion and written in standard English?

Reviewer #1: Yes

Reviewer #3: Yes

6. Review Comments to the Author

Reviewer #1: 1- I am still unclear about the damage model they have used. If you refer to the ABAQUS documentation or any article (https://abaqus-docs.mit.edu/2017/English/SIMACAEMATRefMap/simamat-c-johnsoncook.htm), you can see if they have used a damage initiation and damage evolution model. The damage evolution depends on a displacement/length parameter.

2- In Figure 8b and 8c, there is an element distortion problem. The element may have been deformed by 300% of its original size, and it should have been removed much earlier.

3- Spalling usually occurs in very high-strength steels with a large thickness. When spalling occurs, fragments are produced on the backside of the plate. I do not see such fragments in Figure 9. It appears that the projectile may have completely perforated the plate.

Reviewer #3: Authors can Include latest journals related ballistic behavior of friction stir welded aluminum joints in the introduction

7. PLOS authors have the option to publish the peer review history of their article (what does this mean?). If published, this will include your full peer review and any attached files.

Reviewer #1: No

Reviewer #3: No

---

## [Author Response · Author response to Decision Letter 1]

2 Apr 2023

02.04.2023

Dear Prof. (Dr.) Editor-in-chief,

Thank you for considering the manuscript entitled, “Experimental and numerical investigation of an AA5083 friction stir welded targets against multiple projectile impact: A comparative analysis”, for the publication in PLOS ONE. I am grateful to you and the reviewers for the valuable suggestions provided. I like to resubmit our revised version of the manuscript by adding the response to all your comments. Below please find the answers and actions taken to address these comments. All the suggestions are incorporated and highlighted in BLUE COLOUR in the manuscript. 

NOTE: All the necessary changes/added sentence has been shown in the BLUE COLOUR.

The locations of these changes have been mentioned, where possible, in the action points that respond to each reviewers’ comments. Here are the responses to the reviewer comments:

AUTHOR RESPONSE TO REVIEWER AND EDITOR COMMENTS

Manuscript ID: PONE-D-22-32018R1

Paper title: Experimental and numerical investigation of an AA5083 friction stir welded targets against multiple projectile impact: A comparative analysis

The manuscript has been thoroughly modified and improved the quality of the content to meet the standards of the Journal. All the suggestions made by the learned referees are included in the revised manuscript. We are extremely thankful to the referees & editor(s) for their constructive comments and appreciation. 

Response to Reviewer’s Comments

The authors are grateful to the reviewers for their suggestions which have all contributed to improving the manuscript. Once again, the authors are extremely thankful for the observations and the comments of the reviewers. All the comments are appropriately addressed and now the quality of the article has been appreciably enhanced before the consideration for publication. The rebuttal file is enclosed indicating the revisions incorporated in the article as suggested. The revisions are carried out in BLUE COLOUR in the text of the manuscript for better visibility to the reviewers as well as to the editor. We have made the modifications as per their suggestions in the revised manuscript and changes are also marked up using the “BLUE COLOUR” function.

All in all, the authors should thank the reviewers for their meticulous observations in reviewing the article. All the issues raised by the authors are appropriately addressed as stated in the following table,

Author Response to Reviewer’s Queries

We express our sincere thanks to the reviewers and editor for their valuable suggestions to improve our manuscript. The following are the detailed author’s response (AR) for reviewer’s queries (RQ).

Reviewer #1: Queries and authors response

RQ:1 I am still unclear about the damage model they have used. If you refer to the ABAQUS documentation or any article (https://abaqus-docs.mit.edu/2017/English/SIMACAEMATRefMap/simamat-c johnsoncook.htm), you can see if they have used a damage initiation and damage evolution model. The damage evolution depends on a displacement/length parameter.

AR:1 The Johnson cook damage initiation and damage evolution model depends on following parameters [23-25]

1. Equivalent plastic strain, 2. D1 – D5 are material parameters 3. Strain-hardening constant 4. Thermal-softening constant 5. stress triaxiality ratio.

In addition, the study utilized the ABAQUS finite element code to conduct numerical simulations on a 3D structural FE model of AA5083 target's behavior upon projectile impact. The projectile was modeled as a rigid object, while the target was modeled as a deformable object. The study utilized the Johnson-Cook model to characterize the material behavior of the AA5083 target under the impact loading, and the damage model was used to determine the failure of the target when the damage parameter, D, reached unity. The study used mesh convergence studies to establish the best mesh size for the current situation. The results of the study showed that the target's damage parameters increased with an increase in the projectile's velocity, which ultimately led to the target's failure.

Furthermore, the numerical analysis of the impact of a projectile on a target was conducted using ABAQUS software. The target was modeled as a deformable object, and the projectile was modeled as a rigid object. The Johnson-Cook Model was used to model the material behavior of the target, considering factors such as strain hardening, plastic flow, thermal softening, and fracture. The failure of the target occurs when the damage parameter, D, reaches unity. Mesh convergence research was conducted to establish the best mesh size for the analysis. The model for the aluminum plate was built using around 9345 C3D8R hexahedral elements, and the model for the bullet was built with 2207 tetrahedral elements.

RQ:2 In Figure 8b and 8c, there is an element distortion problem. The element may have been deformed by 300% of its original size, and it should have been removed much earlier.

AR:2 As a result of this finding, there was clear deformation of the plate against the projectile penetration. This was the reason why we took the deformation image for so long, since it had been a while.

RQ:3 Spalling usually occurs in very high-strength steels with a large thickness. When spalling occurs, fragments are produced on the backside of the plate. I do not see such fragments in Figure 8. It appears that the projectile may have completely perforated the plate.

AR:3 We sincerely thank the reviewer for identifying the error that was made by us during the identification of ballistic failure mechanism. The type failure that occurred in the Figure 8 was ductile hole failure. The above correction has been updated in the revised manuscript. 

Reviewer #2: Queries and authors response

RQ:1 Authors can Include latest journals related ballistic behavior of friction stir welded aluminum joints in the introduction.

AR:1 As per reviewer comment, few of the latest journals related to ballistic behavior of friction stir welded aluminum joints has been included in the introduction.

A Scientific explanation of the obtained results has been refined and ameliorated up to a fervent extent. Results are enumerated, test methods are utterly described, and interpretations have been correlated with results and previous literature findings. The overall summary should indicate the progress of the research and its limitations. 

Note: All the necessary changes/added sentence has been shown in BLUE COLOUR.

Thank you very much in advance for taking your time in reviewing this manuscript. 

Sincerely, we hope you will find our revision satisfactory.

Thanks, in anticipation.

Regards,

Dr. Shubham Sharma

(Corresponding author)

---

## [Decision Letter · Decision Letter 2]

19 Apr 2023

Comparative Analysis of Experimental and Numerical Investigation on Multiple Projectile Impact of AA5083 Friction Stir Welded Targets

PONE-D-22-32018R2

Dear Dr. Eldin,

We’re pleased to inform you that your manuscript has been judged scientifically suitable for publication and will be formally accepted for publication once it meets all outstanding technical requirements.

Kind regards,

Antonio Riveiro Rodríguez, PhD

Academic Editor

PLOS ONE

Reviewers' comments:

Reviewer's Responses to Questions

**Comments to the Author**

1. If the authors have adequately addressed your comments raised in a previous round of review and you feel that this manuscript is now acceptable for publication, you may indicate that here to bypass the “Comments to the Author” section, enter your conflict of interest statement in the “Confidential to Editor” section, and submit your "Accept" recommendation.

Reviewer #1: All comments have been addressed

Reviewer #3: All comments have been addressed

2. Is the manuscript technically sound, and do the data support the conclusions?

Reviewer #1: Yes

Reviewer #3: Yes

3. Has the statistical analysis been performed appropriately and rigorously? 

Reviewer #1: N/A

Reviewer #3: Yes

4. Have the authors made all data underlying the findings in their manuscript fully available?

Reviewer #1: Yes

Reviewer #3: Yes

5. Is the manuscript presented in an intelligible fashion and written in standard English?

Reviewer #1: Yes

Reviewer #3: Yes

6. Review Comments to the Author

Reviewer #1: The authors have correctly answered the questions raised, so I consider the paper ready for publication.

Reviewer #3: (No Response)

7. PLOS authors have the option to publish the peer review history of their article (what does this mean?). If published, this will include your full peer review and any attached files.

Reviewer #1: No

Reviewer #3: No

---

## [Editor Report · Acceptance letter]

19 Jul 2023

PONE-D-22-32018R2 

Comparative Analysis of Experimental and Numerical Investigation on Multiple Projectile Impact of AA5083 Friction Stir Welded Targets 

Dear Dr. Eldin:

I'm pleased to inform you that your manuscript has been deemed suitable for publication in PLOS ONE. Congratulations! Your manuscript is now with our production department. 

Kind regards, 

on behalf of

Dr. Antonio Riveiro Rodríguez 

Academic Editor

PLOS ONE